
# A new photometric ozone reference in the Huggins bands: the absolute ozone absorption cross section at the 325 nm HeCd laser wavelength

Christof Janssen[1], Hadj Elandaloussi[1], and Julian Gröbner[2]

[1]LERMA-IPSL, Sorbonne Universités, UPMC Univ. Paris 06, CNRS, Observatoire de Paris, PSL Research University, F-75005 Paris, France
[2]Physikalisches Meteorologisches Observatorium Davos, World Radiation Center, Davos Dorf, Switzerland

*Correspondence to:* C. Janssen (christof.janssen@upmc.fr)

**Abstract.** The room temperature (294.09 K) absorption cross section of ozone at the 325 nm HeCd wavelength has been determined under careful consideration of possible biases. At the vacuum wavelength of 325.126 nm, thus in a region used by a variety of ozone remote sensing techniques, an absorption cross section value of $\sigma = 16.470 \cdot 10^{-21}$ cm$^2$ was measured. The measurement provides the currently most accurate direct photometric absorption value of ozone in the UV with an expanded

(coverage factor $k = 2$) uncertainty $u(\sigma) = 31 \cdot 10^{-24}$ cm$^2$, corresponding to a relative level of two per mil. The measurements are most compatible with a relative temperature coefficient $c_T = \sigma^{-1}\partial_T\sigma = 0.0031$ K$^{-1}$ at 294 K. The cross section and its uncertainty value have been obtained from a generalised linear regression with correlated uncertainties. It will serve as a reference for ozone absorption spectra required for the long-term remote sensing of atmospheric ozone in the Huggins bands. The comparison with commonly used absorption cross section data sets for remote sensing reveals a possible bias of about

2 %. This could partly explain a 4 % discrepancy between UV and IR remote sensing data and indicates that further studies will be required to reach the accuracy goal of 1 % in atmospheric reference spectra.

## 1 Introduction

High resolution reference data for ozone absorption in the UV are widely called for, as this region is used for remote and in-situ measurement of atmospheric ozone concentrations and new measurements are therefore under way in the frame of the ESA

TROPOMI/Sentinel 5 precursor mission that aim at establishing an improved atmospheric spectroscopy database (SEOM-IAS). The demands for increased quality of these atmospheric measurements have been raising continuously over the last decades in order to fulfil the requirement of reliably detecting small atmospheric changes. This has been highlighted in the last report of the "Absorption Cross-Sections of Ozone" (ACSO, http://igaco-o3.fmi.fi/ACSO) joint initiative of the International Ozone Commission (IO3C), the World Meteorological Organisation (WMO) and the Integrated Global Atmospheric Chemistry

Observations (IGACO) O₃/UV subgroup, which was dedicated to studying, evaluating, and recommending the most suitable cross-section data to be used in atmospheric ozone measurements (Orphal et al., 2016). Remote sensing of tropospheric ozone by joint retrieval of UV and IR satellite instruments is another emerging application (e.g. Cuesta et al., 2013) which strongly





depends on unbiased UV spectroscopic data as most of the ozone resides in the stratosphere, but accurate knowledge of the ozone spectrum is also required for the retrieval of other, less abundant trace gases that absorb in spectral ranges where ozone acts as an interfering species.

Reference cross section values with an uncertainty of 1 % or better at the 90 %-confidence level have only recently become available at and around the Hg line position of 253.65 nm (Viallon et al., 2015). This wavelength is particularly important, because absorption at this position is currently used as an ozone standard via standard reference photometers (Hearn, 1961; Viallon et al., 2006). At other wavelengths, such SI traceable data at a similar accuracy level are not available and currently used absorption cross section data in the atmospheric remote sensing of ozone (GSWCB, BDM, BP which stand for Gorshelev, Serdyuchenko, Weber, Chehade and Burrows (Gorshelev et al., 2014; Serdyuchenko et al., 2014), Brion, Daumont and Malicet (Brion et al., 1993; Daumont et al., 1992; Malicet et al., 1995), and Bass and Paur (Bass and Paur, 1985; Paur and Bass, 1985)) do not provide the same level of accuracy and traceability, which might lead to inconsistent and biased results.

However, the UV range between 302 and 340 nm in the Huggins bands of ozone is particularly interesting for ozone column measurements from the ground using Brewer and Dobson spectrophotometers or DOAS ground-based or satellite instruments. The traceability of total column ozone including a comprehensive uncertainty budget is thus an important objective of the Joint Research Project ATMOZ (traceability for ATMospheric total column OZone) within the European Metrology Research Programme (EMRP). The retrieval of total column ozone from solar radiation measurements in the Huggins band requires cross sections with very low uncertainties and well defined temperature coefficients to take into account the effective ozone temperature which varies depending on location and season.

In this article we present new measurements of the UV absorption cross section at the HeCd laser wavelength using the photometric method. Particular attention has been paid to the pressure measurement, the sample purity and to the decomposition of ozone during the measurement process. This led to an improvement of a factor of about ten in the overall uncertainty of the measurement when compared to the reference of Hearn. The measurement thus provides a new reference in the spectral region that is most important for atmospheric remote sensing of ozone. An uncertainty budget following the *ISO Guide to the Expression of Uncertainty in Measurement* (GUM) is given and instrumental biases that might have affected earlier measurements are discussed in detail.

## 2 Experimental setup and methodology

### 2.1 Ozone production and handling

Ozone is produced from high purity oxygen gas (99.9995 %, Air Liquide, France) in a dedicated vacuum system that has been described elsewhere (Janssen et al., 2011). Here we briefly describe some key points (see Fig. 1). The system is made from Pyrex and equipped with all glass valves using PTFE fittings. The only metal parts are gas flasks, pressure gauges, the turbo-molecular pump and stainless steel parts that connect to these components. Ozone is produced by electric discharge at $LN_2$ temperatures in a 3 l reactor, to which copper electrodes are attached at the outside of the walls. After several evaporation and re-condensation cycles, the sample is transferred into a cold trap operating at 65 K, where it is further purified and then



released into the absorption cell. The total volume of the cell, which can be closed off by an all glass stopcock equipped with PTFE fittings, is $113\,cm^3$.

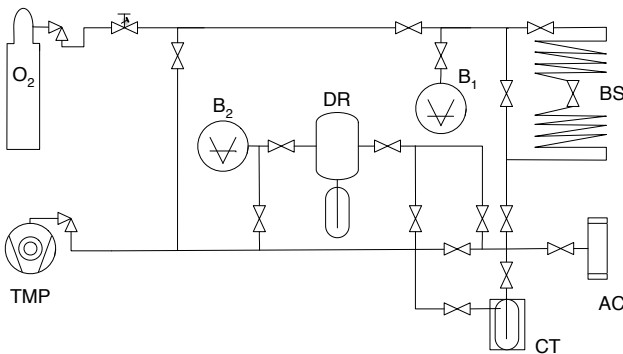

**Figure 1.** Vacuum system for ozone sample preparation. AC – UV absorption cell, $B_1$ – Baratron 690 (10 Torr), $B_2$ – capacitive pressure gauge (1000 hPa), BS – gas buffer spiral, CT – cold trap operable at 65 K, DR – electric discharge reactor chamber (3 l), TMP – turbo-molecular pump.

### 2.1.1 Sample pressure

Over the last three years, the capacitive 10 Torr pressure head (Baratron 690, MKS) of high accuracy (0.08 % nominal) has
been regularly calibrated at 1 year intervals by the French metrology laboratory LNE (last certificate no P156207/1). Due to metal surfaces in the gauge and the stabilisation at $+45\,°C$ , slight ozone decomposition has been observed. In order to improve the stability during the pressure reading, a buffer gas technique has been employed (Janssen et al., 2011).

### 2.1.2 Sample temperature

Four thin film 4-wire Pt100 sensors were distributed over the length of the absorption cell and attached to its outside. The
signals have been registered continuously by a Picotech (pt-104) data logger. Probes and data-logger were calibrated right after the measurement series by an in-house comparison with a traceable standard platinum reference thermometer (SPRT-5626, Hart Scientific) coupled to a readout unit (1502A, Hart Scientific). The calibration uncertainty ($k = 2$) of 14 mK is smaller than observed temperature gradients.

## 2.2 Photometer setup

The absorption measurements are performed using a custom-made photometer, of which an overview is given in Fig. 2. As a light source, a HeCd laser (Kimmon) is used. It delivers around 12 mW of output power at the laser wavelength of 325 nm. The laser light passes a chopper, which modulates the beam amplitude at a frequency of about 2 kHz. The beam is then widened and only a small portion is selected by a $\sim 1$ mm pinhole. A 30:70 beam splitter divides the beam and projects the reflected part on





the reference detector. The transmitted beam is guided twice through the 30 cm long absorption cell using a flat mirror at the backside of the cell. Both, the signal ($I$) and the reference intensities ($I_r$) are measured using cooled Si-photovoltaic detectors (Newport/Oriel) with integrated transimpedance-preamplifiers. The cell windows have a vertical inclination of 3º with respect

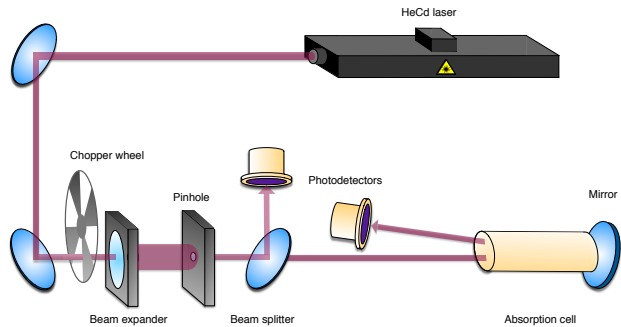

**Figure 2.** Scheme of the optical system. The beam of a HeCd laser is chopped, widened and directed through a pinhole before impinging on a beam splitter. The reflected beam provides the reference signal for the correction of laser intensity fluctuations. The transmitted beam passes the absorption cell twice before being registered by a photodetector. Cell windows are slightly inclined with respect to the optical axis.

to the optical axis in order to avoid light being reflected back and forth between the two cell windows to fall onto the detectors.

The amplitude modulation of the beam intensity allows for phase sensitive detection of the reference and absorption signals, which are measured by digital lock-in amplifiers (SRS 830). Their output signals are registered by a PC using a multi-purpose data acquisition card (NI PCI-6281).

## 2.3   Sample purity and control measurements

In order to control and assess the purity of the ozone sample, a strict protocol of sample preparation, cell filling and pressure
measurements has been followed, as described elsewhere (Janssen et al., 2011). After the measurement has been completed, the sample was re-condensed in a cold trap kept at a temperature of about 65 K. From the residual pressure, the mole fraction $\nu_{nc}$ of non-condensable impurities, such as air, that might have entered the system through small leaks, or oxygen that originates from ozone decomposition, could be estimated. The small mole fraction $\nu_c$ of condensable impurities that might be present in the current absorption cell has been estimated previously (Janssen et al., 2011). No attempt was made to repeat that quantification
here. This was motivated by the fact that ozone decomposition rates in the absorption cell and the amount of non-condensable impurities after the experiment have not changed since. In the earlier study, the mole fractions of water, carbon dioxide, nitrous oxide and nitrate had been measured and found to be $-0.10(17)$, $0.07(7)$, $0.3(3)$, and $-0.01(6)$ mmol mol$^{-1}$, respectively. Moreover, an upper limit of all nitrogen containing impurities of 1.3 mmol mol$^{-1}$ had been found (Janssen et al., 2011, Table I).



## 2.4 Straight-line fit and data evaluation

The Lambert Beer law implies a proportionality between the optical density $\tau$ and the absorbers' column density $\xi = n \cdot L$:

$$\tau = -\ln\left(\frac{(I/I_r)_m}{(I/I_r)_0}\right) = \sigma \cdot (n \cdot L) = \sigma\xi, \tag{1}$$

with the absorption cross section $\sigma$ being the proportionality constant. The $I/I_r$ ratios designate intensities that are normalised

for laser intensity fluctuations by means of a reference beam ($I_r$) and indices $m$ and $0$ indicate an ozone and an empty cell measurement, respectively. In a plot of the optical density $\tau$ versus $\xi$, the cross section $\sigma$ is obtained as the slope of this linear relation. Due to uncertainties in both variables, a standard least squares fit is not appropriate. Because ozone column data are correlated (Bremser and Hässelbarth, 1998; Viallon et al., 2015), a weighted total least squares (WTLS) fit with correlated uncertainties is required. The solution of the total least squares problem ultimately goes back to Deming (1943)

and there is now a rich literature on the York–Williamson algorithm which treats the straight-line adjustment with and without correlated uncertainties in $x$-$y$ data pairs (York, 1966, 1968; Williamson, 1968; York et al., 2004; Reed, 2015, for example). The algorithm is frequently used in environmental, geochemical and isotope studies (e.g. York, 1968; Ludwig and Titterington, 1994; Cantrell, 2008; Wehr and Saleska, 2017). It seems, however, that fewer studies (e.g. Amiri-Simkooei et al., 2014; Bremser and Hässelbarth, 1998; Bremser et al., 2007; Malengo and Pennecchi, 2013) are devoted to the problem when the structure of

the covariance matrix is more complex and when correlations exist between uncertainties in different values of $x$ and/or $y$. This type of question arises in chemometric or metrological applications when calibration lines need to used or when instruments are to be compared.

In order to treat the latter problem, we use here an algorithm from Amiri-Simkooei et al. (2014), which we have implemented using the Mathematica software (Wolfram Research, Inc., 2016). Our implementation provides the fit coefficients $a$ and $b$ of the

straight line function $y = ax + b$, the associated standard uncertainties $u(a)$ and $u(b)$, Pearson's correlation coefficient $r(a,b)$ and the chi-squared value $\chi^2$. The code has been tested on all benchmarks in the ISO technical specification (ISO, 2010, data given in Tables 4, 6, 10, 22 and 25 therein). These include the case of uncertainties in $x$ and $y$, and the two cases when there are covariances associated with the $y$ values and when covariances are associated with both, the $x$ and the $y$ values. Our results agreed within all digits indicated. We also note that our implementation further matched all example calculations given in the

original publication of Amiri-Simkooei et al. (2014). This comprises the classical data set from Pearson with York's weights assuming no correlations in $x$-$y$ data pairs (York et al., 2004), but agreement to all digits as given by Reed (2015) is also obtained when such a correlation is considered.

## 3   Analysis and uncertainty budget

### 3.1   Laser wavelength

We are not aware of direct interferometric measurements of the $4d^9 5s^2 : {}^2D_{3/2} \to 4d^{10}5p : {}^2P^\circ_{1/2}$ laser transition at 325 nm. Reported wavelength values are based on the analysis of emission spectra of the $Cd^+$ ion produced in electric discharges.




Previous atmospheric studies (Lakkala et al., 2008; Lantz et al., 2002) using a HeCd laser mostly report an air wavelength of 325.029 nm. This number emanates probably from term energies reported in the handbook of basic atomic spectroscopic data (Sansonetti and Martin, 2005) that are ultimately based on one comprehensive study (Shenstone and Pittenger, 1949). Other data bases, reference tables and handbooks (e.g. Reader et al., 1980; Haynes, 2015) recommend the somewhat different

value of 325.033 nm, which corresponds to the wavelength that Shenstone and Pittenger (1949) have actually measured for this transition. While the 0.004 nm difference between the measured and the term energy derived transition energy is compatible with the measurement uncertainty of $\sim 0.1\,\mathrm{cm}^{-1}$, Burns and Adams (1956) have confirmed the previously measured value at a much lower degree of uncertainty ($< 0.01\,\mathrm{cm}^{-1}$). Indeed, their measurement resulted in a vacuum wavelength $\lambda_{vac} = 325.126$ nm, which under standard conditions ($T = 15°\mathrm{C}$, $p = 101325$ Pa) and reasonable variation of the air molecular

composition ($RH = (50\pm50)\,\%$ and $x(\mathrm{CO_2}) = (0.4\pm0.1)\,\mathrm{mmol\,mol^{-1}}$) corresponds to the air wavelength $\lambda_{air} = 325.033$ nm with all figures being significant (Ciddor, 1996).

Further confidence into the claimed wavelength accuracy might be obtained by comparing measured and tabulated wavelengths of the well studied HeNe laser. The ASD database gives an air wavelength of 632.8614 nm for the Ne I transition, corresponding to a vacuum wavelength $\lambda_{\mathrm{HeNe}} = 632.9914$ nm (when standard conventions are applied: $T = 15°\mathrm{C}$, $p = 101325$ Pa,

$x(\mathrm{CO_2}) = 0.33\,\mathrm{mmol\,mol^{-1}}$, $RH = 0$). This result agrees to all digits with the reproducible line position of typical HeNe lasers (Mielenz et al., 1968).

## 3.2 Optical density

The stability of the laser fluctuation corrected signal $I/I_r$ is shown in Fig. 3. The displayed curve is characteristic for our system and allows for a conservative uncertainty estimation, because curves at other measurement days gave values at a lower

level. We have chosen integration times of about 30 s, for the ozone and empty cell measurements, that were taken within a time span of about 2 minutes. For the ratio $(I/I_r)_m / (I/I_r)_0$ we infer a measurement uncertainty of

$$u(\tau) = 1.8 \cdot 10^{-4} \sqrt{(3 + \exp(\tau))/2}, \qquad (2)$$

where $\tau$ denotes the optical density of the absorption measurement. In deriving the above expression we have assumed that firstly the measurements of the empty cell (0) and of the cell filled with ozone (O$_3$) are stochastically independent (white

noise behaviour) during the whole measurement period that lasted for about 2 minutes, that secondly intensities of the empty cell and the reference beam contribute equally and that thirdly the relative uncertainty of the filled cell signal scales with $1/\sqrt{I_m} \sim \sqrt{\exp(\tau)}$. The measurement signal being the ratio of two intensity ratios (thus the product/ratio of four measurement signals), the relative uncertainty must yield $\sqrt{2}$ times the 30 s level in Fig. 3 for $\tau = 0$. Note that the value of $2.1 \cdot 10^{-4}$ in Fig. 3 is conservative, because the spectrometer stability at other days has always been better. We have thus replaced that number by

the moderately lower and more representative value of $1.8 \cdot 10^{-4}$. For optical densities between 0.025 and 0.32, as in this study, Eq. (2) implies standard uncertainties $u(\tau)$ between $2.6 \cdot 10^{-4}$ and $2.7 \cdot 10^{-4}$. This is only slightly higher than the residual scatter ($2.5 \cdot 10^{-4}$) of our measurements (see Fig. 5b in Sect. 4.1). The uncertainties correspond to relative values $u_r(\tau)$ being in the 0.08 to 1.0 % range.



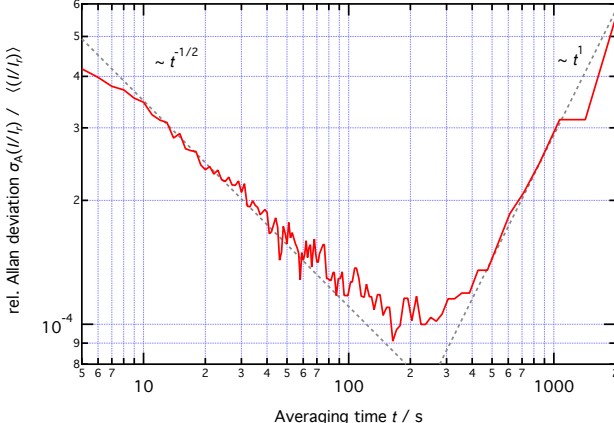

**Figure 3.** The stability of the photometer has been checked each measurement day after 2 h of laser warm up and before the measurements were performed. The measurement showing the lowest level of stability is shown. Data obtained on other days fall below the above curve, which is characterised by a white noise dependence ($\sim t^{-1/2}$) for about 100 to 180 s before a linear drift component becomes dominant.

### 3.3 Temperature

A temperature gradient along the cell of about 100 mK has been observed. In the absence of more accurate data, we have determined the cell temperature as the average of the minimal and maximal temperature during a measurement ($\sim 30$ s). Taking into account the uncertainty of the calibration (7 mK), the standard uncertainty has been determined as

$$u^2(T) = (T_\mathrm{max} - T_\mathrm{min})^2/12 + (7\,\mathrm{mK})^2. \tag{3}$$

### 3.4 Pressure

The capacitive pressure sensor has been regularly calibrated at the french national metrology institute LNE. The calibration determines the measurement uncertainty from the scatter of repeated readings and the standard uncertainty of the LNE working standard. In the relevant 10 Torr range, reading errors have been shown to be negligible. However, there is a small pressure rise observed during the measurement, which is likely due to some ozone decomposition. This 0.04 Pa rise leads to an additional standard uncertainty of 0.01 Pa. Taken together with the calibration uncertainty we obtain

$$u(p) = 0.05\,\mathrm{Pa} + 10^{-3}p. \tag{4}$$

The laboratory temperature has an effect on offset and span of our sensor. While the offset is always readjusted, we need to consider the manufacturer specified span temperature coefficient of $2 \cdot 10^{-5}/°\mathrm{C}$. We are always within $\pm 2.5$ K of the calibration temperature, which adds an uncertainty of $5 \cdot 10^{-5}$ to the span. Because this is at least 20 times smaller than the calibration uncertainty, we can simply neglect it here. The dominant pressure uncertainties in Eq. (4) are of type B and do not reduce





**Table 1.** Major contributions ($> 99\,\%$) to the standard uncertainty in the optical path length. Other factors contribute less than $1\,\%$.

| No | Parameter | Probability distribution[a] | Value $x$ / mm | Standard uncertainty $u(x)$ / µm | Sensitivity coefficient $c(x)$ | Contribution $|c(x)u(x)|$ / µm |
|---|---|---|---|---|---|---|
| 1. | Window thickness $d$ | rect. | 2.0 | 57.7 | $-4.008$ | 231 |
| 2. | Beam[b] ordinate on entrance window $y_1$ | rect. | $-2$ | 577 | 0.0601 | 35 |
| 3. | Beam[b] ordinate on exit window $y_2$ | rect. | $-3$ | 577 | $-0.0598$ | 35 |
| 4. | Shortest window distance[c] $\mathcal{L}$ | gauss. / rect. | 301.77 | 10.2 | 2.003 | 20 |
| | Optical path length $L$ | | 596.654 | 243 | | |

[a] rect. – rectangular, gauss. – Gaussian,

[b] incident and reflected beam, which have the same $y$-coordinates, contribute equally. Each entry must thus be accounted for twice.

through repeated measurements. This is confirmed by the long-term drift between our sensor and the LNE working standard, which shows a characteristic pattern that evolves slowly over the calibration period of three years time.

## 3.5 Optical path length

The path length has been determined from the window thickness and calliper measurements of the outer cell dimensions
combined with the observation of the entrance and exit positions on the two cell windows. A HeNe laser beam has temporarily been superposed to the UV beams and the cell centre axis in order to determine the different inclination angles. The procedure is described in detail in Appendix A. Altogether, seventeen different measurands contribute to the determination of the cell geometry and the orientation of the two beams with respect to the cell. All of these are included in the uncertainty budget of the optical path length $L = l_1 + l_2 = (596.654 \pm 0.243)\,\text{mm}$ (see Table 1), which is obtained as the sum of the individual lengths
on the round trip through the cell. We only list and discuss the four factors that contribute most. The remaining, non-listed quantities add in by less than $1\,\%$. The most important ($95\,\%$) contribution to the uncertainty is from the window thickness. The manufacturer specified tolerance, which we verified on other windows of the same production batch, is $\pm 0.1\,\text{mm}$. We therefore deduce a standard uncertainty $u(d) = 100\,\text{µm}/\sqrt{3} = 57.7\,\text{µm}$. Superposing transparent millimetre paper on the cell windows allowed to determine the coordinates ($x_1, y_1, x_2, y_2$, see Appendix 3.5) where the laser beams passed the cell windows.
A standard uncertainty of $u(y_1) = u(y_2) = u(x_1) = u(x_2) = 1\,\text{mm}/\sqrt{3} = 577\,\text{µm}$ is estimated for these measurements. They impact the length measurement at second and third place. Since the cell is passed by the laser beam two times, resulting in two beams with separate optical path lengths, these two contributions need to be accounted for twice. Due to the window inclination, the length is more sensitive to the vertical coordinate.

In fourth place comes the shortest distance $\mathcal{L}$ between the inclined, but not exactly parallel windows. It has been measured
using a calliper that has been compared to gauge block combinations with overall lengths of 290 and 300 mm. The resolution of the calliper being $10\,\text{µm}$, the comparison with the gauge blocks always gave perfect agreement. The uncertainty of the length



measurement has therefore been obtained as quadratic sum of two contributions: the standard uncertainty related to the calliper resolution ($5\,\mu m/\sqrt{3}$) and the error of the mean of eight measurements at two different days, which was found to be $9.8\,\mu m$.

The finite dimensions of the laser beam have also been taken into account and found to be negligible compared to other factors. The beam divergence has been estimated using the divergence angle $\alpha_d \sim 5 \cdot 10^{-4}$, determined from beam profile measurements before the entrance and after the exit of the cell. To first approximation, its effect ($\sim \sec(\alpha_d) - 1 \simeq \alpha_d^2/2 \simeq 10^{-7}$) on the path length is negligible. The finite diameter has been explored through numerical simulations of parallel displacements of our beam centre. We shifted the centre by $\pm 1\,mm$ (the beam diameter is between 2.6 and 3.3 mm) in one direction and found that the average of the two displacement is within the length of the beam centre by less than 2 parts in $10^6$.

### 3.6 Sample purity

The sample purity is characterised by the mole fractions of condensable ($\nu_c$) and non-condensable gases ($\nu_{nc}$). In a previous study, the mole fractions of $CO_2$, $H_2O$, $N_2O$, and $NO_3$ have been determined and an upper limit for the sum of all oxides and hydrogen oxides of nitrogen, of $\nu_{nc} < 1.3\,mmol\,mol^{-1}$ had been found. Despite the fact that the observed mole fractions of the directly measurable quantities were all within one standard uncertainty or close to 0, we assume a rectangular probability distribution function (pdf) with bounds at 0 and 1.3 mmol mol$^{-1}$. We thus obtain $\nu_{\mathrm{HxCyOz}} = 0.65\,mmol\,mol^{-1}$ with a standard uncertainty of $0.38\,mmol\,mol^{-1}$. When combining this result with the observations on $CO_2$ and $H_2O$, we obtain $\nu_c = 0.62\,mmol\,mol^{-1}$ with a standard uncertainty $u(\nu_c) = 0.42\,mmol\,mol^{-1}$.

The mole fraction of non-condensables has been determined from measurements of the residual pressure after condensation of the cell content as

$$\nu_{nc,\mathrm{max}} = \gamma \cdot p_{res}/p, \tag{5}$$

where $p_{res}$ is the residual gas pressure, and $\gamma = 7.24 \pm 0.44$ a factor which takes into account the volume ratios and temperature gradients between the absorption cell, the volume where the residual pressure measurements have been made and the cold finger where ozone is frozen back. The uncertainty of $\gamma$ comprises the reproducibility of test measurements (1.9 %) and varying levels of $LN_2$ that change the effective volume of the cold finger (5.8 %). Residual pressure measurements are impacted by the thermal transpiration effect (Daudé et al., 2014) caused by the heating of the gauge (45 °C). It can be taken into account by assuming that the actual pressure is somewhere between the indicated value and the maximum of 4.2 % induced by thermal transpiration. This leads to a +2.1 % correction of the pressure reading with an associated standard uncertainty of the residual pressure measurement of 1.2 %.

The value in Eq. (5) is a limiting value, as it is has been obtained only after the measurement and it is likely that the non-condensables not only enter into the measurement cell during sample admission, but especially when the re-condensation of ozone is made and the residual pressures are measured. In the absence of further information, we simply assume a rectangular probability distribution $0 \leq \nu_{nc} \leq \nu_{nc,\mathrm{max}}$ for each measurement. $\nu_{nc,\mathrm{max}}$ varies between 1.3 and $4.8\,mmol\,mol^{-1}$ with an average of 2.9 mmol mol$^{-1}$. This amounts to a typical value of $\nu_{nc} = 1.4\,mmol\,mol^{-1}$ with a standard uncertainty of $u(\nu_{nc}) =$





0.84 mmol mol⁻¹. We note that the dominating source of uncertainty comes from the unknown origin of the residual pressure and not from individual measurements.

### 3.7 Temperature dependence

For small variability, the temperature dependence of the absorption cross section $\sigma$ in the vicinity of some reference value $\sigma_0$ at $T = T_0$ is given by

$$\frac{\sigma(T)}{\sigma_0} - 1 = c_T (T - T_0) \tag{6}$$

where $c_T$ is the normalised linear temperature coefficient. As shown further below, our measurements are most compatible with $c_T = 0.0031\,\mathrm{K}^{-1}$. This is close to data in the literature: $0.0033\,\mathrm{K}^{-1}$ (Serdyuchenko et al., 2014), $0.0042\,\mathrm{K}^{-1}$ (Malicet et al., 1995), and $0.0039\,\mathrm{K}^{-1}$ (Paur and Bass, 1985) at 325.03 nm and 294 K. Since the absorption cross section in the Huggins band is strongly wavelength and temperature dependent, we prefer using $c_T = 0.0031\,\mathrm{K}^{-1}$ as other values might be biased by small wavelength shifts. For the uncertainty estimate we assume a rectangular pdf with $0.0011\,\mathrm{K}^{-1}$ half width to obtain $u(c_T) = 0.00064\,\mathrm{K}^{-1}$.

### 3.8 Uncertainty budget for a single measurement

The uncertainty of a cross section measurement is obtained from the Beer-Lambert law (Eq. (1)), taking into account that the ozone column density $\xi_i$ is given by

$$\xi_i = (n \cdot L)_i = (1 - \nu_c - \nu_{nc,i})\, L \frac{p_i}{k_{\mathrm{B}} T_i}, \tag{7}$$

where the different quantities have their previously defined meanings. However, as will become clear later, it is useful to define an adjusted ozone column density

$$x_i = \xi_i \left(1 + c_T (T_i - T_0)\right) = \left(1 + c_T (T_i - T_0)\right) \left(1 - \nu_c - \nu_{nc,i}\right) L \frac{p_i}{k_{\mathrm{B}} T_i}, \tag{8}$$

where the slight temperature dependency of the absorption cross section is incorporated into the coordinate axis (see Sect. 4.1). Quantities with added index $i$ vary between runs and must be determined for each individual ozone absorption measurement, while others, such as the path length $L$, remain always the same. These constants necessarily introduce a correlation between different values of $x_i$. For an individual measurement, where we neglect correlations and the temperature dependency of the cross section, simple error propagation rules yield the following equation for the relative uncertainty of the cross section

$$u_r^2(\sigma) = u_r^2(\tau_i) + u_r^2(\xi_i) = u_r^2(\tau_i) + u_r^2(k_{\mathrm{B}}) + u_r^2(L) + u_r^2(p_i) + u_r^2(T_i) + \frac{u^2(\nu_c) + u^2(\nu_{nc,i})}{\left(1 - u(\nu_c) - u(\nu_{nc,i})\right)^2}. \tag{9}$$

The different contributions are summarised in Table 2 and a total relative standard uncertainty of $u_r(\sigma) = 2.3 \cdot 10^{-3}$ is obtained for an individual measurement. This, for the moment, neglects the uncertainty caused by repeating measurements at slightly different temperatures, taken into account in the full analysis presented later in Sect. 4. The most prominent contributions





**Table 2.** Uncertainty budget of a single absorption cross section measurement at average pressure

| Parameter | Unit | Probability distribution[a] | Typical or recommended value $X$ | Rel. standard uncertainty $u_r(X)$ |
|---|---|---|---|---|
| Length $L$ | mm | rect. | 596.654 | $4.1 \cdot 10^{-4}$ |
| Mole fraction complement of non-condensable impurities $(1 - \nu_{nc})$ | 1 | rect. | $1 - 1.4 \cdot 10^{-3}$ | $8.2 \cdot 10^{-4}$ |
| Mole fraction complement of condensable impurities $(1 - \nu_c)$ | 1 | rect. | $1 - 6.2 \cdot 10^{-4}$ | $4.2 \cdot 10^{-4}$ |
| Temperature $T$ | K | gauss. | 294.09 | $5.8 \cdot 10^{-4}$ |
| Pressure $p$ | hPa | gauss. | 7.6 | $1.1 \cdot 10^{-3}$ |
| Opt. density $\tau$ | 1 | gauss. | 0.18 | $1.6 \cdot 10^{-3}$ |
| Temperature dependence of cross section[b] $c_T$ | $K^{-1}$ | rect. | 0.0031 | $2.1 \cdot 10^{-1}$ |
| Boltzmann constant $k_B$ | $J\,K^{-1}$ | – | $1.38064852 \cdot 10^{-23}$ | $5.7 \cdot 10^{-7}$ |
| Cross section $\sigma$ | $cm^2$ | | $16.47 \cdot 10^{-21}$ | $2.3 \cdot 10^{-3}$ |

[a] rect. – rectangular, gauss. – Gaussian

[b] contributes through additional weighting factor $\Delta T / T \sim 1.9 \cdot 10^{-3}$

$(> 1\text{‰})$ are due to the measurement of the optical density and the pressure. Repeated measurements will allow to improve on the measurement uncertainty, provided that correlations in the pressure and other data contributing to the ozone column density are taken into account.

### 3.9 Correlations between realisations of the ozone column density

Eq. (8) provides also the basis for the evaluation of measurement correlations. Constants in that equation clearly introduce a correlation between different values of $x_i$, but individual realisations of temperature, pressure and the mole fraction of non-condensable impurities are also not strictly independent from one run to another, because their measurements rely on the same calibrations and sensors. The correlation coefficients $r_{ij} = u^2(x_i, x_j) / (u(x_i)u(x_j))$ between two measurements $i$ and $j$ of the ozone column $x$ can be calculated from Eq. (8), using correlations between the independent measurement quantities and a

generalised error propagation rule. Details of the procedure are presented in Appendix B. We obtain

$$
\frac{u^2(x_i, x_j)}{x_i x_j} = u_r^2(k_B) + u_r^2(L) + \frac{(T_i - T_0)(T_j - T_0)u^2(c_T)}{(1 - c_T(T_i - T_0))(1 - c_T(T_j - T_0))} + \frac{u^2(\nu_c) + u^2(\nu_{nc,i}, \nu_{nc,j})}{(1 - \nu_c - \nu_{nc,i})(1 - \nu_c - \nu_{nc,j})} + \frac{u^2(p_i, p_j)}{p_i p_j}
$$
$$
+ \frac{(1 - c_T T_0)^2}{(1 - c_T(T_i - T_0))(1 - c_T(T_j - T_0))} \frac{u^2(T_i, T_j)}{T_i T_j}. \tag{10}
$$

where $k_B$, $L$, $\nu_c$ and $c_T$ are the independent quantities common to all determinations and where the variables $p_i$, $T_i$ and $\nu_{nc,i}$ are newly determined in each run. The similarity with Eq. (9) is apparent. Indeed with the exception of the term for the optical

density, we immediately recover Eq. (9) by setting $i = j$ and $u(c_T) = c_T = 0$. There is no uncertainty $u(T_0)$ associated to the arbitrarily chosen reference temperature $T_0$, which explains the absence of a corresponding term. The calculation of the different terms for $i \neq j$ is detailed in the remainder of this section.





**Table 3.** Contributions to correlation coefficients between different realisations $x_i$ and $x_j$ of the ozone column ($i \neq j$).

| Quantity ($y$) | First order contribution | Sensitivity coefficient $(\partial_y x_i)(\partial_y x_j)/(x_i x_j)$ | | | $r_{ij}$ | |
|---|---|---|---|---|---|---|
| | | Average | Min | Max | Min | Max |
| Boltzmann constant $k_{\mathrm{B}}$ | $u_r^2(k_{\mathrm{B}})$ | $3.25 \cdot 10^{-13}$ | – | – | – | – |
| Length $L$ | $u_r^2(L)$ | $1.67 \cdot 10^{-7}$ | – | – | – | – |
| Temperature coefficient $c_T$ | $(T_i - T_0)(T_j - T_0)u^2(c_T)$ | $-4.79 \cdot 10^{-9}$ | $-5.00 \cdot 10^{-7}$ | $4.23 \cdot 10^{-7}$ | – | – |
| Mole fraction $\nu_c$ of condensable impurities | $u^2(\nu_c)$ | $1.76 \cdot 10^{-7}$ | $1.76 \cdot 10^{-7}$ | $1.76 \cdot 10^{-7}$ | – | – |
| Mole fraction $\nu_{nc,i}$ of non-condensables | $u(\nu_{nc,i}, \nu_{nc,j})$ | $2.26 \cdot 10^{-7}$ | $5.01 \cdot 10^{-8}$ | $6.36 \cdot 10^{-7}$ | – | – |
| Pressure $p_i$ | $u(p_i, p_j)(p_i p_j)^{-1}$ | $4.70 \cdot 10^{-8}$ | $1.90 \cdot 10^{-8}$ | $2.89 \cdot 10^{-7}$ | – | – |
| Temperature $T_i$ | $u(T_i, T_j)(T_i T_j)^{-1}$ | $5.00 \cdot 10^{-11}$ | $5.00 \cdot 10^{-11}$ | $5.00 \cdot 10^{-11}$ | – | – |
| Modified ozone column density $x_i$ | | | | | 0.76 | 1.00 |

The Boltzmann constant and the absorption path length contribute via their absolute or relative standard uncertainties to $r_{ij}$. We obtain (see Tables 2 and 3):

$$u_r(x_i)u_r(x_j)r_{i,j}\big|_{k_{\mathrm{B}},L} = 1.672 \cdot 10^{-7}. \tag{11}$$

Similarly, the contribution to the correlation coefficient through temperature variation of the absorption cross section is

$$u_r(x_i)u_r(x_j)r_{i,j}\big|_{c_T} = \frac{4.033 \cdot 10^{-7}\mathrm{K}^{-2}(T_i - T_0)(T_j - T_0)}{(1 - c_T(T_i - T_0))(1 - c_T(T_j - T_0))}. \tag{12}$$

As discussed in Sect. 3.8, individual measurements of non-condensable impurities ($\nu_{nc,i}$) are essentially fully correlated, which is due to the fact that it is not known whether the small amounts of residual gases have already been present during the measurement or were added only afterwards. We thus have $u^2(\nu_{nc,i}, \nu_{nc,j}) = u(\nu_{nc,i})u(\nu_{nc,j})$ and, if we add the constant contribution from the condensables $u^2(\nu_c) = 1.764 \cdot 10^{-7}$, we obtain:

$$u_r(x_i)u_r(x_j)r_{i,j}\big|_{\nu_{nc},\nu_c} = \frac{0.577^2 u(\nu_{nc,i})u(\nu_{nc,j}) + 1.764 \cdot 10^{-7}}{(1 - \nu_c - \nu_{nc,i})(1 - \nu_c - \nu_{nc,j})}. \tag{13}$$

Temperature measurements are assumed to be non-correlated except for the contribution due to sensor calibration (7 mK):

$$u_r(x_i)u_r(x_j)r_{i,j}\big|_{T} = \frac{49 \cdot 10^{-6}\,\mathrm{K}^2(1 - c_T T_0)^2}{(1 - c_T(T_i - T_0))(1 - c_T(T_j - T_0))T_i T_j} \tag{14}$$

The uncertainty of the pressure measurement is essentially limited by the calibration. Repeated measurements at the same pressure will thus be fully correlated. Less is known about the correlation of measurements at different pressures. As pointed

out by Viallon et al. (2015), assuming a high degree of correlation does not alter the derived value of the absorption cross section, but leads to a conservative uncertainty estimate. Therefore we assume full correlation $u^2(p_i, p_j) = u(p_i)u(p_j)$:

$$u_r(x_i)u_r(x_j)r_{i,j}\big|_{p} = u_r(p_i)u_r(p_j). \tag{15}$$

The range of values for the different contributions is indicated in Table 3. Taking all parts together one gets hold of the correlation coefficient $r_{ij}$. Evidently, $r_{i,j} = 1$ for $i = j$, but we still find an average value of $\overline{r_{ij}} = 0.94$ for $i \neq j$, indicating

a very strong correlation between different measurements of the ozone column density. Most of this is due to the foreign gas contamination and the optical path length.





## 4  Analysis and results

Figure 4 shows the results of 27 individual measurements and an unweighted linear fit to the data. The measurements span the range of $\tau$ between 0.025 and 0.32, corresponding to ozone columns from 0.15 to $1.95 \cdot 10^{19}$ cm$^{-2}$. A high coefficient of determination ($r^2 = 0.999977$) attests to the excellent linearity between optical densities $\tau$ and ozone columns $\xi$. Before the cross section value can be derived, the impact of temperature and the choice of the fitting model need to be examined.

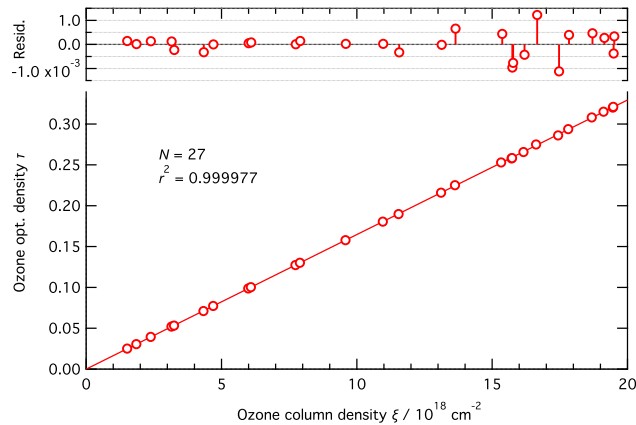

**Figure 4.** Unweighted linear fit to 27 individual pairs of ozone optical ($\tau$) and column ($\xi = n \cdot L$) densities.

### 4.1  Preliminary analysis and fit model

As discussed previously, the data were obtained for temperatures varying slightly in the range between 293.17 and 295.37 K. This leads to some scatter due to the temperature dependence of the absorption cross section. Using a local linear dependency on temperature, the optical density will be given by

$$\tau(\xi, T) = \sigma(T)\xi = \sigma_0 \left[ 1 + c_T (T - T_0) \right] \xi = \sigma_0 x(T), \tag{16}$$

where we have defined the new variable $x = [1 + c_T (T - T_0)]\xi$ (see Eq. 8). In order to determine the cross section at the average temperature $T_0$, we can now plot $\tau$ vs $x$, which directly yields the cross section $\sigma_0$ as the slope term. While we allow for an offset $a$ in the linear fit that serves as an additional control, we also need to explore the possibility of non-linearities in our measurement chain, possibly caused by a saturation of the detectors or by other effects in the electronic acquisition and
15  amplification modules. This can be accomplished by including a quadratic term $(b\xi^2)$ in the fit, leading to the following model:

$$\tau(\xi) = a + \sigma_0 \left[ 1 + c_T (T - T_0) \right] \xi + b\xi^2. \tag{17}$$

Fig. 5 shows the residuals of fitting this function for different scenarios. In the lower panel relatively large residuals with a reduced standard deviation of $S_r = S \cdot \sqrt{27/25} = 5.0 \cdot 10^{-4}$ and prominent features at $\xi \sim 1.7 \cdot 10^{19}$ cm$^{-2}$ are observed, when



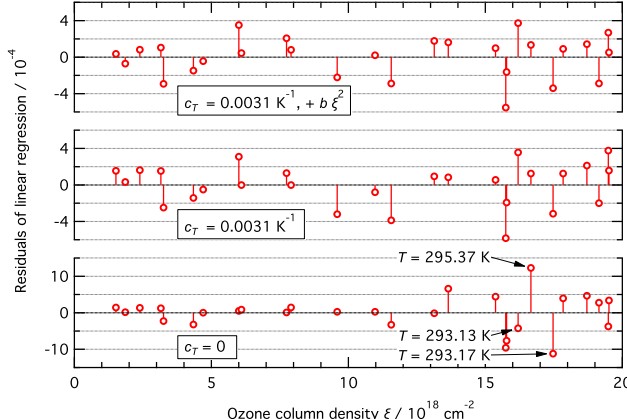

**Figure 5.** Study of residuals in the fitting of the absorption data using different variants of Eq. (17) as fit models. From bottom to top, the number of free fit parameters increases. The result of a simple linear fit neglecting the temperature dependence of the absorption cross section ($c_T = 0$ in Eq. (17)) is shown on the bottom. Residual values with highest and the two lowest sample temperatures are indicated. The same fit including a temperature dependence ($c_T = 0.0031\,\mathrm{K}^{-1}$) is displayed on the middle panel. The top panel figures residuals when the fit includes an additional quadratic term ($+b\xi^2$) in the ozone column density. Scales in the two upper graphs are enlarged by a factor of 2.5.

we assume $b = c_T = 0$. Interestingly, the most variable temperature conditions (between $-0.92$ and $+1.28\,\mathrm{K}$ with respect to the average) prevailed during measurements at these column densities. When a first order correction $c_T = 0.0031\,\mathrm{K}^{-1}$ for the temperature is taken into account, the largest residual features disappear (as shown in the middle panel of Fig. 5) and the spread of residuals (max$-$min) is reduced by a factor of 2.0. Correspondingly, the reduced standard deviation of the temperature

corrected residuals in the middle panel of Fig. 5 of $S_r = S \cdot \sqrt{27/24} = 2.5 \cdot 10^{-4}$ is only half of that in the lower panel. It has to be noted that this number is only slightly lower than the standard uncertainty of the optical density $\tau$, derived in Sect. 3.2.

Allowing for a quadratic term $+b\xi^2$ in the fit affects residuals (shown on the top panel of Fig. 5) only marginally, diminishing the reduced standard deviation just by 3 % to yield $S_r = 2.4 \cdot 10^{-4}$. At the same time, the quadratic term introduces a strong anti-correlation between fit parameters ($r(\sigma_0, b) = -0.98$). This indicates that, while the effect of temperature on the fit is well

significant, quadratic terms are not. Our restriction to a straight-line fit is thus well justified in what follows. We also fix the temperature coefficient to our best fit value of $c_T = 0.0031\,\mathrm{K}^{-1}$, because the value is consistent previously observed data (see Sect. 3.7) and because reasonable changes to this parameter do not modify our result significantly (see Sect. 4.2).

## 4.2 Linear regression

After having established the fitting model, the data are evaluated using the weighted total least squares algorithm with correlated

$x$-$\tau$ data. Table 4 summarises the results of the analysis. The cross section $\sigma_0 = 1.6470 \cdot 10^{-20}\,\mathrm{cm}^2$ with a relative standard uncertainty $u_r(\sigma) = 9.3 \cdot 10^{-4}$ ($k = 1$) is obtained. The small offset within the uncertainty range indicates that the data comply with our hypothesis of a straight-line passing through the origin, thus that our measurement follows the Beer-Lambert law. The





**Table 4.** Linear fit statistics including standard uncertainty ($k = 1$) values.

| Quantity | Unit | Value |
|---|---|---|
| Degrees of freedom $\nu$ | 1 | 25 |
| Slope $\sigma$ | $10^{-20}\,\text{cm}^2$ | 1.64704 |
| Offset $\tau_0$ | 1 | $6.684 \cdot 10^{-6}$ |
| Slope uncertainty $u(\sigma)$ | $10^{-23}\,\text{cm}^2$ | 1.530 |
| Offset uncertainty $u(\tau_0)$ | 1 | $1.033 \cdot 10^{-4}$ |
| Pearsons coefficient $r(\sigma, \tau_0)$ | 1 | $-0.4092$ |
| Chi-squared $\chi^2$ | 1 | 20.8469 |

$\chi^2$ value falls within the 10 and 90 % quantiles of the cumulative $\chi^2_{25}$-distribution, which also indicates that the straight-line hypothesis does not need to be rejected and that the uncertainty analysis is compatible with our data.

The importance of considering covariances in this type of photometric absorption measurements (Bremser et al., 2007) is once more emphasised by comparing our results with numbers obtained when these covariances are omitted. Ignoring covariances firstly leads to an unrealistically small value of $\chi^2_{25}$ (12.6 instead of 20.8), and, secondly underestimates $u(\sigma)$ by 34 %. However, the absolute value of $\sigma$ is remarkably robust against the neglect of covariances (and changes only by 0.011 %). This finding is in line with the discussion of Viallon et al. (2015), where an effectively constant correlation coefficient $r = r_{ij}$ for all $x_i$-$x_j$ pairs ($i \neq j$) has been assumed. But we suspect that this might not generally be true. In particular if $r_{ij}$ strongly varies as a function of $i$ and $j$, we expect that the value of the cross section changes too upon considering covariances. A possible scenario would be a measurement where different pressure sensors are utilised in different pressure ranges, possibly leading to little correlation between low and high pressure values, while maintaining a high correlation coefficient between measurements using the same gauge. We also note that the result is de facto independent of our choice of $c_T$. Using one of the highest value reported in the literature so far ($c_T = 0.0042\,\text{K}^{-1}$ (Paur and Bass, 1985) instead of $c_T = 0.0031\,\text{K}^{-1}$), the derived cross section value changes by less than 1 part in $10^5$ and the uncertainty estimate is not at all affected.

## 5 Discussion

### 5.1 Comparison with laboratory data

Table 5 compares our result with previously published high resolution data or cross sections commonly used for atmospheric retrieval. For convenience, the Table not only provides the analysis of the original data, but also the recent parametrisation of cross section data from BP, BDM and GSWCB and their uncertainties provided by Weber et al. (2016). Their investigation agrees well (better than 0.2 %) with our analysis of the original data, when cross sections are smoothed. Interestingly, all of the literature data only insignificantly deviate from our reference measurement. Except for BP, the literature data sets agree well with each other at 325 nm, but they show values about 2 % higher than our measurement, independent whether they have been





**Table 5.** Comparison of absolute high resolution absorption cross section data of ozone at 325.126 nm in vacuum (325.033 nm in air). Smoothed data have been obtained from applying a Savitzky-Golay filter of order 2 over the range of 0.1 nm.

| Data set[a] | Temperature[b] | Cross section $\sigma$ ($10^{-21}$ cm$^2$) | | | Rel. standard uncertainty[c] $u_r(\sigma)$ (%) | | Relative deviation[d] |
|---|---|---|---|---|---|---|---|
| | (K) | original | smoothed | Weber et al. (2016) | from paper | Weber et al. (2016) | from *this work* (%) |
| BP (1985) | 294.1 | 16.335 | 16.335 | 16.315 | 2.3 | 2.31 | $-0.8 \, / \, 0.7 \ldots 2.5$ |
| BDM (1995) | 294.1 | 16.864 | 16.863 | 16.896 | $2-4$ | 1.74 | 2.4 |
| VOPB (2001) | 294.1 | 16.855 | 16.819 | – | $4-7$ | – | 2.1 |
| GSWCB(2014) | 294.1 | 16.716 | 16.740 | 16.735 | $1.1-3$ | 1.65 | 1.6 |
| *This work* | 294.09 | 16.470 | – | – | 0.093 | | |

[a] Data were obtained from the ACSO website: igaco-o3.fmi.fi/ACSO/cross_sections.html. References are: BP – Bass and Paur (1985); Paur and Bass (1985), BDM – Daumont et al. (1992); Malicet et al. (1995), VOPB – Voigt et al. (2001), and GSWCB – Gorshelev et al. (2014); Serdyuchenko et al. (2014)

[b] The temperature dependence of the literature data has been taken into account using a quadratic parametrisation $\sigma(t) = \sigma_0(1 + c_1 t + c_2 t^2)$, where $t = T - 273.15$ K. BP and GSWCB have provided corresponding coefficients $\sigma_0$, $c_1$ and $c_2$. For BDM and VOPB, these have been obtained from a quadratic fit to cross sections given at fixed temperatures.

[c] The uncertainty estimation of BDM contains the effect of wavelength shifts, not considered by BP, GSWCB and VOPB.

[d] Based on smoothed data. BP cross sections suffer from wavelength bias: neg. value uncorrected; range of pos. values after correction (see text).

calibrated at the Hearn (1961) value (VOPB), or not (GSWCB, BDM). This is different from the situation around the top of the Hartley band. In that region Viallon et al. (2015) have observed that data can be divided in two distinct groups: one, where values have been scaled to the absolute absorption cross section of Hearn, including the old Bremen (VOPB) data, and one where the absolute scale has been determined independently, such as their own measurements, the new Bremen (GSWCB) and

the Reims (BDM) data, the former group giving values about 2 to 3 % higher than the latter. Obviously this is not the case at the HeCd laser wavelength (see Table 5 and Fig. 6).

The negative offset of the BP data must mostly be explained by a wavelength bias (see Fig. 6). Early evaluations report shifts of the BP cross sections between 0.025 and 0.05 nm (Malicet et al., 1985, 1995; Orphal, 2003) and the existence of this bias is confirmed by comparison with atmospheric spectra (Orphal et al., 2016). When compared to atmospheric spectra, the

BDM data, however, do not require any shift. In calculating differential cross sections (Platt and Perner, 1984) in the 320 to 330 nm range, we determine shifts of 0.049 nm between BP and BDM and 0.23 nm between BP and GSWCB. In combination with the strong wavelength dependency in the Huggins band (Fig. 6), which is quite different from the peak of the Hartley band that essentially is a spectrally flat region, the wavelength shift leads to a systematic bias in the cross section value. Using a linear variation of the cross section of $\partial\sigma/\partial\lambda = -1.1 \cdot 10^{-20}$ cm$^2$ nm$^{-1}$ (at 325.1 nm) common to all wavelength dependent

measurements in Fig. 6, the BP cross section at the reference wavelength would actually be higher than our measurement by 0.7 to 2.5 % (assuming the BP wavelength bias to be somewhere between 0.023 and 0.049 nm), thus implying a similar cross section offset than the other data sets (see Table 5).

Despite the nominal agreement of our determination with all other measurements listed in Table 5, a concern might be the fact that all of these take higher values (when wavelength shifts are corrected for). It must therefore be pointed out that the study

of Hearn (1961) gives values that are consistently lower than BDM and GSWCB by 2.7 to 3.7 % at three Hg line wavelength



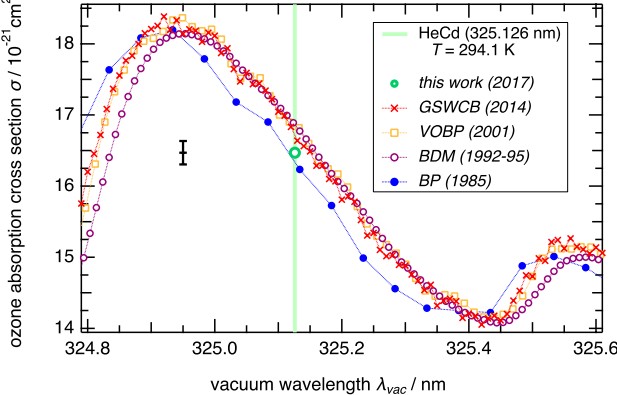

**Figure 6.** Comparison of high resolution ozone absorption data around the 325 nm HeCd laser line (position indicated by vertical bar). The vacuum wavelength is used on the ordinate scale. Data are from the same sources as in Table 5. Straight lines between measurement points were inserted for visual guidance. Individual uncertainty bars have been omitted from the graph. The uncertainties of this work are smaller than the symbol size. Uncertainty assignments for other spectra are given in Table 5. As a reference, the black vertical bar indicates the $\pm 1\%$ relative uncertainty range at $1.647 \cdot 10^{-20} \, \mathrm{cm}^2$.

positions $(289.4 - 302.2 \, \mathrm{nm})$ in the region around $300 \, \mathrm{nm}$. Furthermore, neither GSWCB nor BDM mention any particular precaution against multiple reflections in their optical setups. The presence of such reflections within the absorption cell leads to an overestimation of the absorption cross section (Viallon et al., 2006). Under the specified conditions, we estimate that a corresponding bias between $+0.3\%$ and $+1.2\%$ for the GSWCB data or between $+0.3$ and $+0.8\%$ for the BDM cross section could exist.

## 5.2 Atmospheric implications

The discrepancy at the HeCd laser wavelength indicates a $2\%$ room temperature bias in current atmospheric reference spectra used by a variety of remote sensing platforms and techniques (Brewer, Dobson, LIDAR, Umkehr, SBUV, TOMS, OMI, SCIAMACHY and GOME(-2), see Orphal et al. (2016), for example). If that bias applies to a larger wavelength region $(\sim 310 - 340 \, \mathrm{nm})$ and to most of the atmospheric temperature range, actual retrievals in this spectral region systematically underestimate atmospheric ozone by about $2\%$. Although identical in magnitude this tentative bias in the Huggins bands is different from the ongoing discussion whether the reference absorption cross section of Hearn (1961) at the Hg line position of $253.65 \, \mathrm{nm}$ in the Hartley band should be reduced by about $2\%$ (Viallon et al., 2015; Orphal et al., 2016), because both the GSWCB and the BDM data are already compatible with the lower value at $253.65 \, \mathrm{nm}$, and only the BP dataset that is not any more recommended for atmospheric retrieval (Orphal et al., 2016) would be affected by the revision of the absorption cross section at $253.65 \, \mathrm{nm}$.





There is a long standing consistency problem of atmospheric ozone derived from remote sensing in UV and IR spectral regions (e.g. Barbe et al., 2013; Janssen et al., 2016). Both, laboratory (Picquet-Varrault et al., 2005; Gratien et al., 2010; Guinet et al., 2010) and atmospheric (Kagawa et al., 2007; Viatte et al., 2011) studies imply that using recommended spectroscopic data in the UV (BP/BDM) and IR (HITRAN2012, Rothman et al. (2013)) lead to results that disagree by about 4 to 5 %,

with ozone abundances inferred from IR measurements being higher. Thus comparing measured ($msd$) and database ($db$) IR-intensities ($I$) with UV cross sections ($\sigma$) through the ratio

$$\widetilde{R} = (I/\sigma)_{msd} / (I/\sigma)_{db} = (\sigma_{db}/\sigma_{msd})(I_{msd}/I_{db}), \tag{18}$$

the above studies indicate a value $\widetilde{R}$ around 1.04 or 1.05, whereas consistent data require $\widetilde{R} = 1$. Note that the factor $(I/\sigma)_{msd}$ in this equation is the ratio of two absorption signals, where factors except for the molecular parameters, such as the concentra-

tion or light path geometry ideally cancel. Our new absorption cross section at 325 nm suggests that the currently used values of $\sigma_{db}$ might be too high by about 2 % ($\sigma_{db}/\sigma_{msd} \sim 1.02$) which brings IR and UV results to within 2 or 3 % if this bias is taken into account. The remaining discrepancy is already close to most measurement uncertainties, but also agrees remarkably well with the value $I_{msd}/I_{db} - 1 = 2.5\%$ observed by Guinet et al. (2010), who investigated 15 intense lines at $8.8\,\mu$m in the $\nu_1$ fundamental. This situation thus is similar to the spectral conditions in the atmospheric UV-IR comparison of Viatte et al.

(2011) using Brewer and FTS instruments, their IR analysis being based on the $\nu_1$-$\nu_3$ region at $9.6\,\mu$m. The atmospheric comparison of Kagawa et al. (2007) between concentrations from TOMS (UV) and from ground based FTS (IR) doesn't directly depend on the intensities in the $\nu_1$ fundamental. But the fact that most atmospherically relevant ozone vibrational intensities in HITRAN directly depend on transition moments of the $\nu_1$ and $\nu_3$ fundamentals, implies that IR intensities in the $3\,\mu$m region should be corrected by the same amount (Rothman et al., 2005; Flaud et al., 2003), implying that the discrepancy observed in

their study would be resolved at the same time.

The new ozone cross section at the HeCd laser wavelength thus not only provides the first reference value with sub-percent accuracy for ozone spectra in the Huggins bands, it also supplies independent evidence for a shared contribution of IR and UV biases to the UV-IR consistency problem of atmospheric ozone. This is the first evidence directly based on a measurement in the Huggins band, i.e. in the same UV band that is actually utilised for the atmospheric (Kagawa et al., 2007; Viatte et al.,

2011) and laboratory (Picquet-Varrault et al., 2005; Gratien et al., 2010) inter-comparisons. A previous laboratory study (Guinet et al., 2010) depended on UV measurements in the Hartley band. Further systematic temperature and wavelength dependent studies with high accuracy will be required to bear out the possible bias in currently used atmospheric reference spectra (BDM, GSWCB and forthcoming data) and confirm our assertion with respect to the share of bias between UV and IR data in the spectroscopic data bases.

**6 Conclusions**

Using a HeCd laser spectrophotometer we have obtained the currently most accurate measurement of an ozone absorption cross section in the Huggins bands, and in the spectral region used by a variety of remote sensing techniques and platforms. The cross





section $\sigma = (16.470 \pm 0.031) \cdot 10^{-21} \, \text{cm}^2$ was found at $\lambda_{vac} = 325.126 \, \text{nm}$ and a full uncertainty budget in accordance with the guide to expression of uncertainty in measurements (GUM) has been presented. The expanded ($k = 2$) relative uncertainty is at the 2 permil level and thus significantly below the accuracy of previous measurements and well below the current target of 1 % for atmospheric applications. This high accuracy level has been made possible by the use of a special ozone production

and handling system and an elaborate analysis of the light path in a cell with slightly non-parallel windows. The measurement, together with a recent study at several wavelengths ($244 - 257 \, \text{nm}$) in the Hartley band (Viallon et al., 2015), demonstrates that a sub-percent accuracy can now well be achieved in laboratory ozone absorption investigations and promises that the accuracy of atmospheric measurements can be improved significantly.

Our new reference value suggests that absorption spectra currently used for atmospheric remote sensing of ozone possibly

need to be revised towards lower values in the Huggins bands by about 2 %. Such a revision would likely impact most ozone retrievals in the UV and would also reduce the $\sim 4, \%$ UV-IR discrepancy reported in atmospheric and laboratory studies by a factor of 2. The remaining 2 to 3 % need to be attributed to a bias in the IR data, which is compatible with a previous independent IR study. The often cited target uncertainty of 1 % has obviously not yet been reached in atmospheric reference spectra. This implies that further studies are required. The possible bias in the atmospheric reference spectra is likely wavelength dependent,

because atmospheric reference spectra need to be acquired in spectral slices to be combined to cover all the range from the UV to the NIR, which is a consequence of the seven orders of magnitude in absorption between the Hartley and the Wulff bands. One would thus ideally make high accuracy measurements at regular wavelength intervals (10 or 20 nm, or so) in order to investigate this wavelength dependence. Unfortunately, this is not easily feasible due to the need of suitable laser sources at all these different wavelengths. As a next step, we propose to extend the current measurements to selected UV and VIS

wavelengths (particularly around 254, 325 and 633 nm, for example) using both, gas and tuneable lasers as well as to include the whole temperature range down to 190 K. In this way, relevant reference points or even small regions for actual or new atmospheric reference spectra can be obtained. These can be used to calibrate existing and future cross section data, to assess their accuracy, to identify wavelength shifts and to assure traceability in limited wavelength regions.

## Appendix A: Path length in cell with non-parallel windows

Here we describe how the the absorption path length in a cell with inclined (but not necessarily parallel) windows is obtained. The full analytic expression has been derived using an algebraic software package (Mathematica) and our uncertainty analysis has been based on this exact solution. Its analytical form is too clumsy to be fully reproduced here. We prefer to give the closed analytic solution for a cell with parallel windows, together with the first order correction for slightly non-parallel windows.

The general situation with arbitrarily inclined window plates is illustrated in Fig. A1. In the laboratory system, we define the

$z$-axis along the centres of two parallel plates of thickness $d$ and radius $R$, measured between the outer surfaces of the windows. The first center is located at $z = 0$, the second at $z = L_0$. $x_1$ and $y_1$ coordinates, respectively, designate axes in the vertical and horizontal directions in the laboratory frame ($x_2$ and $y_2$ are similarly defined at the origin of the second window). Because windows are assumed to be spherical, two Euler angles suffice to define the window inclination: $\beta_1$ for rotation around the





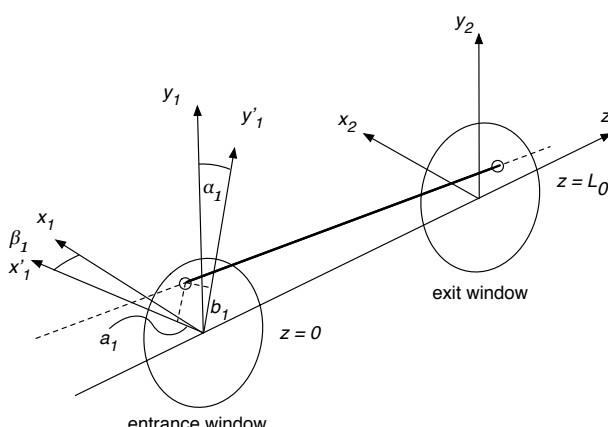

**Figure A1.** Geometry for the calculation of the absorption length. Two arbitrarily oriented spherical windows are located at a distance $L_0$ along the z-axis. Unprimed coordinates designate systems in the laboratory frame and primed coordinates are on the window surfaces. Euler angles $\alpha_1$, $\beta_1$ The light beam, indicated by a bold line between the windows, passes through the entrance window at point $(x'_1, y'_1) = (a_1, b_1)$ on the outer surface. The window thickness is neglected in the drawing.

$y_1$-axis, $\alpha_1$ for rotation around the newly obtained $x'_1$-axis. The passage of the light beam is defined by the coordinates on the entrance $((x'_1, y'_1) = (a_1, b_1))$ and exit window $((x'_2, y'_2) = (a_2, b_2))$ surfaces. We define $\alpha = (\alpha_1 + \alpha_2)/2$ and $\beta = (\beta_1 + \beta_2)/2$ to be the average inclination angles and $\Delta y = b_2 - b_1$ and $\Delta x = a_2 - a_1$ the changes of the horizontal and vertical displacements of the window coordinates between the beams' exit and entrance. We can also characterise the degree of non-parallelness by

introducing the angle differences $\Delta\alpha = (\alpha_2 - \alpha_1)/2$ and $\Delta\beta = (\beta_2 - \beta_1)/2$.

Let us first note that the window center distance $L_0$ can be obtained from the shortest distance $\mathcal{L}$ between the two inclined plates, measured with a calliper where the two outside jaws are oriented along the $y_1$ and $y_2$-axes :

$$L_0 = \frac{1}{\cos(\alpha - \Delta\alpha)} \left( \mathcal{L}\cos^2\alpha + \mathcal{L}\sin(\alpha)\sin(\alpha - 2\Delta\alpha) + R\sin(2\Delta\alpha) \right). \tag{A1}$$

In deriving this formula we have made the convenient but non-restricting assumption that $0 \le \alpha_1 \le \alpha_2 < \pi/2$. Assuming that

windows are parallel ($\Delta\alpha = \Delta\beta = 0$), the length of a single pass is given by

$$l^{(0)} = l_p^{(0)} \mathcal{A}(\Delta x, \Delta y), \quad \text{where} \quad l_p^{(0)} = \frac{\mathcal{L}}{\cos\alpha} \left( 1 - \frac{2d}{\mathcal{L}\cos\beta} \right) \tag{A2}$$

is the length of the beam propagating parallel to the $z$-axis and where the correction term

$$\mathcal{A}(\Delta x, \Delta y) = \sqrt{1 + \cos^2\alpha \left[ \left(\frac{\Delta x}{\mathcal{L}}\right)^2 + \left(\frac{\Delta y}{\mathcal{L}}\right)^2 - 2\frac{\Delta x}{\mathcal{L}}\sin\beta - 2\frac{\Delta y}{\mathcal{L}}\tan\alpha \right]} \tag{A3}$$

takes into account any beam inclination with respect to the $z$-axis. We note in passing, that $L_0 = \mathcal{L}\sec\alpha$ for parallel windows

and that in this case $l_p^{(0)}$ is just the difference between the outer window distance and twice the effective window thickness.



When, as in our setup, windows are slightly non-parallel ($\Delta\alpha \ll 1$, $\Delta\beta \ll 1$) the single path length might conveniently be expressed as a linear expansion in the non-parallelness parameters. Thus to first order terms, the length may be expressed as

$$l = l^{(0)} + l^{(1)}_{\Delta\alpha}\Delta\alpha + l^{(1)}_{\Delta\beta}\Delta\beta + \mathcal{O}(\Delta\alpha^k \Delta\beta^m), \quad k,m \geq 0 \wedge k+m = 2. \tag{A4}$$

As a matter of fact, the agreement between this approximation and the exact solution is better than $2\,\text{nm}$ for a single pass in our configuration. For extreme conditions with $\alpha = 5°$, $\beta = 0$ or $\alpha = 0$, $\beta = 5°$ and $\Delta\alpha$ and $\Delta\beta$ in the $0.5°$ range, where the beam passes through the $30\,\text{cm}$ cell within $5\,\text{mm}$ of the center, we find that the linear approximation for one pass always agrees with the full analytic solution by better than $12\,\mu\text{m}$, which is close to the calliper resolution. Let us introduce some quantities for deriving the coefficients in Eq. (A4):

$$\bar{x} = x + \Delta x/2, \quad \bar{y} = y + \Delta y/2, \quad \Delta = \sqrt{\Delta x^2 + \Delta y^2} \tag{A5}$$

$$\mathcal{B} = 3\sin\alpha + \sin 3\alpha - 4\frac{R+\bar{y}}{\mathcal{L}}\cos\alpha. \tag{A6}$$

Here, the average horizontal $\bar{x}$ and vertical $\bar{y}$ beam displacements have been introduced. Using these abbreviations and the definition of $\mathcal{A}$ in Eq. (A3), the partial derivatives for the first order corrections in $\Delta\beta$ and $\Delta\alpha$ are given as

$$l^{(1)}_{\Delta\alpha} = \frac{\mathcal{L}}{2\mathcal{A}}\left\{\mathcal{B}\left[\left(1+\frac{2d}{\mathcal{L}\cos\beta}\right)\left(\frac{\Delta x}{\mathcal{L}}\sin\beta\frac{\Delta y}{\mathcal{L}}\tan\alpha\right) - \frac{1}{\cos^2\alpha} - \frac{2d}{\mathcal{L}\cos\beta}\left(\frac{\Delta}{\mathcal{L}}\right)^2\right]\right.$$
$$\left. + 4\cos\alpha\left(1-\frac{2d}{\mathcal{L}\cos\beta}\right)\left(\frac{\Delta y}{\mathcal{L}}-\tan\alpha\right)\left(\frac{\bar{x}}{\mathcal{L}}\sin\beta+\frac{\bar{y}}{\mathcal{L}}\tan\alpha\right)\right\} \tag{A7}$$

$$l^{(1)}_{\Delta\beta} = \frac{2\bar{x}\cos\alpha}{\mathcal{A}}\left\{\cos\beta - \frac{2d}{\mathcal{L}}\left[1-\left(\frac{\mathcal{A}}{\cos\alpha\cos\beta}\right)^2\right]\right\}. \tag{A8}$$

## Appendix B:  Correlation terms

The correlation coefficients $r_{ij} = u^2(x_i, x_j)/(u(x_i)u(x_j))$ can be obtained from a generalised uncertainty propagation rule:

$$u^2(x_i(y), x_j(y)) = \sum_{k=1}^{n}\sum_{r=1}^{m}\left(\frac{\partial x_i}{\partial y_{k,r}}\right)\left(\frac{\partial x_j}{\partial y_{k,r}}\right)u^2(y_{k,r}) + \sum_{\substack{l,k=1\\l\neq k}}^{n}\sum_{\substack{s,r=1\\s\neq r}}^{m}\left(\frac{\partial x_i}{\partial y_{k,r}}\right)\left(\frac{\partial x_j}{\partial y_{l,s}}\right)u^2(y_{k,r}, y_{l,s}). \tag{B1}$$

Summation indices $(k,l)$ and $(r,s)$ respectively go over the number $n$ of observables $y$ in Eq. (8) and the number $m$ of different measurements. By setting $i = j$ and identifying covariance terms $u^2(x_i, x_i)$ by variances $u^2(x_i)$, we recover the familiar propagation rule for standard uncertainties with contributions from both, variance and covariance terms. Eq. (B1) considerably simplifies when cross-correlation terms vanish. In our case, variables $k_B$, $\nu_c$, $L$, $c_T$ and $T_0$ are common to all realisations and stochastically independent of all other quantities. Their covariance terms thus disappear completely. Due to temperature ($T$), pressure ($p$) and residual gas ($\nu_{nc}$) measurements being independent from each other, covariances between $T$ and $p$, between $T$ and $\nu_{nc}$ and between $p$ and $\nu_{nc}$ also mutually vanish. So do the variance terms of these variables, because





their sensitivity coefficients are necessarily 0 for $i \neq j$. One thus finds

$$\frac{u^2(x_i, x_j)}{x_i x_j} = u_r^2(k_{\mathrm{B}}) + u_r^2(L) + \frac{(T_i - T_0)(T_j - T_0)u^2(c_T)}{(1 - c_T(T_i - T_0))(1 - c_T(T_j - T_0))} + \frac{u^2(\nu_c) + u^2(\nu_{nc,i}, \nu_{nc,j})}{(1 - \nu_c - \nu_{nc,i})(1 - \nu_c - \nu_{nc,j})} + \frac{u^2(p_i, p_j)}{p_i p_j}$$
$$+ \frac{(1 - c_T T_0)^2}{(1 - c_T(T_i - T_0))(1 - c_T(T_j - T_0))} \frac{u^2(T_i, T_j)}{T_i T_j}. \tag{10}$$

Note that $u(T_0) = 0$ due to $T_0$ being an arbitrary constant and that we have introduced the normalisation factor $x_i x_j$, which

5  expresses the covariances on the same footing than (the squared) relative uncertainties.

*Author contributions.* CJ and HE designed and performed the experiments. CJ analysed and interpreted the data. CJ and JG wrote the paper.

*Competing interests.* The authors declare no competing interests.

*Acknowledgements.* We thank the entire technical staff at LERMA and the glass blowers J.P. Francois from LPMAA and F. Thibout from
LKB, without whom the work could not have been realised. CJ also would like to thank M. Spidell and G. Nave from NIST for extended
10  discussions, in which they openly shared their expertises on HeCd lasers and atomic emission lines. The work received funding from the
French national programme LEFE/INSU. JG received financial support by the European Metrology Research Programme (EMRP) within
the joint research project EMRP ENV59 ATMOZ "Traceability for atmospheric total column ozone." The EMRP is jointly funded by the
EMRP participating countries within EURAMET and the European Union.





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
