# Peer review of "A new photometric ozone reference in the Huggins bands: the absolute ozone absorption cross section at the 325 nm HeCd laser wavelength"

_Atmospheric Measurement Techniques, 2017_

## Referee Comment (RC1) · Anonymous Referee #2 · 26 Oct 2017

General comments: The purpose of the manuscript concerns "a new photometric ozone reference in the Huggins bands: the absolute ozone absorption cross section at the 325 nm HeCd laser wavelength". The manuscript is well written in a concise way. The state of art for the ozone absorption cross section measurements during the last three decades is very well documented. The authors explain the need for the atmospheric community to have very precise ozone absorption cross sections in the UV range as this region is widely used for remote and in-situ measurements of atmospheric ozone concentration. The focus on the 325 nm HeCd laser wavelength allows a new precise and absolute measurement of ozone absorption cross section at room temperature. As ozone is an unstable gas, many factors may affect the purity and the decomposition of the studied sample. So the experimental setup and methodology are clearly detailed in order to control the ozone sample during all the measurements. Throughout the manuscript, particular attention is paid to the determination of uncertainty budget and all possible instrumental biases. This very precise and detailed experimental work and the used method (fitting model and linear regression) to analyze the data, lead to a new measurement of the ozone absolute absorption cross section at 325 nm of high quality. This new cross section value suggests that absorption spectra used for atmospheric remote sensing of ozone possibly need to be revised as the obtained value is about 2% lower than the previous ones. This is of high interest for the atmospheric community. Moreover, this new value can also be used to calibrate existing and future cross section data. Conclusion: I highly recommend this manuscript for publication.

Specific comments: No specific comments

Technical errors: None

---

## Author Comment (AC1) · 9 Nov 2017

On behalf of all authors, I would like to sincerely acknowledge the work of the anonymous referee #2. We thank him or her for carefully reviewing our manuscript and we refrain from any changes to the manuscript at this point, since no particular issues were raised.

---

## Referee Comment (RC2) · Anonymous Referee #1 · 17 Jan 2018

General comments: As authors point out, there is a demand for accurate ozone absorption cross-section values, and thorough measurements are a prerequisite for reliable reference data. Measurements of the absolute values are particularly challenging for such an unstable gas as ozone, and authors provide strong evidence that their experimental approach allows to achieve very low uncertainty of the resulting data.

The manuscript is well structured and highlights all milestones of the performed research.

Authors provide analysis of experimental data and line up a very detailed uncertainty budget summary.

[Figure]

Measured ozone absorption cross-section value at the 325nm HeCd laser wavelength can be used not only as a reference in the UV, but also serve for further investigation of discrepancy between the UV and IR remote sensing results.

The mentioned 2% difference with respect to existing broadband ozone absorption cross-section datasets can certainly become a motivating factor for further cross-section measurements with unprecedented uncertainties.

Specific comments: none

Technical errors: none

---

## Author Comment (AC2) · 18 Jan 2018

We gladly acknowledge that the referee shares our view on the importance and implication of our measurements. Since no specific or technical comments were raised by the referee, no detailed response or correction is required at this point.

We would like to take the opportunity to once again expressing our thanks to both of the anonymous referees for their work.

---

## Author Response (AR1)

**Author Response**

**January 18, 2018**

We are pleased about the very positive responses to our manuscript submission. Since both referees did not point out any issues, we did not see any need for a point by point reply. Nevertheless, we have made very few technical/orthographic corrections to the first submission, that in no way represent a change to the manuscript content. Changes made are explicitly mentioned in the list herebelow. Most of these changes ghave been detected by the latexdiff software and are thus indicated on the following pages by color. If this is not the case (at two instances), this has been pointed out deliberately.

- 1. The affiliation 1 has changed as the *Université Pierre et Marie Curie* has fusioned to form the new *Sorbonne Université*. This change was not detected by the latexdiff software.
- 2. *parallelness* has been corrected to *parallelism*
- 3. expertises in the acknowledgement has been changed to expertise
- 4. Wulff band has been corrected to Wulf band
- 5. non-condensable has ben changed to noncondensable
- 6. *permil* has been changed to *per mil*
- 7. vs has been changed to versus
- 8. when calibration lines need to used or when instruments are to be compared has been changed into when calibration lines need to be used or when instruments are to be compared
- 9. The three references Bass, A. M. and Paur, R. J. (1985), Malicet, J., Brion, J., and Daumont, D. (1985), and Paur, R. J. and Bass, A. M. (1985) have been corrected in the reference section as to conform with the requirements of the publishing company. This has been signaled to the authors by one of the editors of the conference proceedings. This change has not been detected by the latexdiff software.

With best regards

Christof Janssen

[revised manuscript text omitted]